# Regulatory Effects of Three-Dimensional Cultured Lipopolysaccharide-Pretreated Periodontal Ligament Stem Cell-Derived Secretome on Macrophages

**DOI:** 10.3390/ijms24086981

**Published:** 2023-04-10

**Authors:** Yuran Su, Sifan Ai, Youqing Shen, Wen Cheng, Chenyu Xu, Lei Sui, Yanhong Zhao

**Affiliations:** 1Department of Prosthodontics, School and Hospital of Stomatology, Tianjin Medical University, Tianjin 300070, China; 2Key Laboratory of Bioactive Materials, Ministry of Education, State Key Laboratory of Medicinal Chemical Biology, College of Life Sciences, Collaborative Innovation Center of Chemical Science and Engineering, and National Institute of Functional Materials, Nankai University, Tianjin 300071, China; 3Department of Orthodontics, School and Hospital of Stomatology, Tianjin Medical University, Tianjin 300070, China

**Keywords:** periodontal ligament stem cells, lipopolysaccharide, three-dimensional culture, secretome, macrophages

## Abstract

Phenotypic transformation of macrophages plays important immune response roles in the occurrence, development and regression of periodontitis. Under inflammation or other environmental stimulation, mesenchymal stem cells (MSCs) exert immunomodulatory effects through their secretome. It has been found that secretome derived from lipopolysaccharide (LPS)-pretreated or three-dimensional (3D)-cultured MSCs significantly reduced inflammatory responses in inflammatory diseases, including periodontitis, by inducing M2 macrophage polarization. In this study, periodontal ligament stem cells (PDLSCs) pretreated with LPS were 3D cultured in hydrogel (termed SupraGel) for a certain period of time and the secretome was collected to explore its regulatory effects on macrophages. Expression changes of immune cytokines in the secretome were also examined to speculate on the regulatory mechanisms in macrophages. The results indicated that PDLSCs showed good viability in SupraGel and could be separated from the gel by adding PBS and centrifuging. The secretome derived from LPS-pretreated and/or 3D-cultured PDLSCs all inhibited the polarization of M1 macrophages, while the secretome derived from LPS-pretreated PDLSCs (regardless of 3D culture) had the ability to promote the polarization of M1 to M2 macrophages and the migration of macrophages. Cytokines involved in the production, migration and polarization of macrophages, as well as multiple growth factors, increased in the PDLSC-derived secretome after LPS pretreatment and/or 3D culture, which suggested that the secretome had the potential to regulate macrophages and promote tissue regeneration, and that it could be used in the treatment of inflammation-related diseases such as periodontitis in the future.

## 1. Introduction

Macrophages are innate immune cells that regulate the maintenance of tissue homeostasis, host defense during pathogen infection and tissue repair in response to tissue injury [1]. Macrophages are a group of phenotypic heterogenic cells which are classified into two major subsets: classically activated (M1) macrophages, which are polarized by lipopolysaccharide (LPS), either alone or in combination with interferon (IFN)-γ, and produce pro-inflammatory cytokines such as interleukin (IL)-1β, IL-6, tumor necrosis factor (TNF)-α, etc.; and alternatively activated (M2) macrophages, which are polarized by IL-4 and IL-13 and produce anti-inflammatory cytokines such as IL-10, transforming growth factor (TGF)-β, etc. [2].

Periodontitis is a worldwide chronic multifactorial inflammatory disease associated with dental biofilm and characterized by the progressive destruction of periodontal support tissue, including the periodontal ligament and alveolar bone [3]. The dysregulation between the periodontal microbial community and the host immune inflammatory response is considered to be the leading cause of periodontitis [4], and macrophages play important immune response roles in the occurrence, development, and regression of periodontitis [5]. M1 macrophages regulate osteoclast activation by promoting the Type 1 T helper response, stimulate osteoclast progenitors and secrete a large number of pro-inflammatory cytokines involved in periodontitis progression and bone resorption, while M2 macrophages participate in inflammation resolution and tissue regeneration by secreting anti-inflammatory mediators and adjusting Type 17 T helper (Th17) and regulatory T (Treg) cells functions [6]. The imbalance of M1/M2 macrophages is responsible for periodontal tissue destruction. Adjusting the proportion of macrophages with different polarization phenotypes to regulate the inflammatory response is effective in treating periodontitis.

Mesenchymal stem cells (MSCs) have the ability for self-renewal, multi-directional differentiation and immune regulation, which have been widely studied in the field of regenerative medicine, and they have shown considerable therapeutic potential in a variety of inflammatory immune diseases [7,8]. Earlier studies mainly attributed the therapeutic effect of MSCs to their ability to transplant locally and differentiate into multiple tissues, but recent studies have demonstrated that implanted cells cannot survive for a long time [9]. MSCs exhibit their effects mainly due to the production of a large number of regulatory substances involved in intercellular communication, including cytokines, chemokines, immune regulating factors, growth factors and extracellular vesicles (EVs) in the conditioned medium (CM), collectively known as the secretome of MSCs, which plays important roles in the regulation of key biological processes [10,11,12,13,14,15]. The MSC-derived secretome is a cell-free therapy strategy that can effectively avoid the problems of the direct use of MSCs and provide more advantages over MSC-based applications, such as avoiding potential safety risks associated with cell transplantations and facilitating the evaluation of safety, dosage and potency, as well as collection and storage [8,16].

The immunomodulatory function of MSCs was initially thought to be intrinsic; however, recent studies have shown that it is not constitutive and requires some degree of inflammatory response and/or other environmental stimuli such as hypoxia and the composition of extracellular matrix (ECM), etc. [17,18]. Lipopolysaccharide (LPS) is the main component of the cell wall of Gram-negative bacteria, which plays a key role in the interaction between pathogens and host immune system [19]. LPS pretreatment of MSC-derived EVs significantly reduced the inflammatory response of inflammatory diseases [20,21], including periodontitis [22], by inducing M2 macrophage polarization. In addition, when pretreated with LPS, MSCs displayed increased secretion of IL-6 and IL-8 and retained high expression of these cytokines for over seven days without the influence of cell division [23], indicating that MSCs possessed some characteristics of immune cells that enabled them to retain information from environmental stimulation for a period of time and to have a better therapeutic effect in immunomodulation.

In addition, in order to mimic the MSC niche in vivo, researchers have explored the use of different three-dimensional (3D) cell systems and found that these could improve the immunomodulatory properties of MSCs. Three-dimensional culture of MSC spheroids efficiently produces an immunosuppressive secretome by regulating macrophages [24,25,26,27]. Compared with spherical culture, the secretome secreted by 3D hydrogel-cultured MSCs had higher anti-inflammatory and immunomodulatory properties and greater regeneration potential [28]. In previous studies, we designed Biotin–DFYIGSRGD peptides that could self-assemble into supramolecular hydrogels (termed SupraGel) for 3D cell culture [29]. It is worth noting that the separation of cells and SupraGel was successfully achieved by centrifugation after addition of phosphate-buffered saline (PBS), which reminded us that the SupraGel could be used as a tool to collect the secretome of MSCs.

As previously indicated, the immunomodulatory properties of MSCs are not intrinsic; similarly, the MSC-derived secretome is not a constant mixture of secretory factors but changes depending on the existing microenvironment of the MSCs [30]. Appropriate pretreatment may induce MSCs to secrete a secretome with enhanced regenerative potential [31].

To sum up, secretome derived from LPS-pretreated or 3D-cultured MSCs have enhanced immunomodulatory effects by regulating the polarization phenotypes of macrophages. Further study is needed on the combination and optimization of different factors in MSC pretreatment to better mimic the inflammatory environment in vivo to then further enhance its immunomodulatory effect through macrophage phenotype switching.

As a subset of MSCs, periodontal ligament stem cells (PDLSCs) can differentiate into osteoblasts and fibroblasts and regenerate periodontal ligament-like tissues, as well as have self-renewal and immunosuppressive properties [32].

In this study, we constructed a PDLSC–SupraGel culture system and collected PDLSC-derived secretome after LPS pretreatment and/or 3D culture to verify their regulatory effect on macrophage polarization and migration. We also detected the immune cytokines in the secretome to speculate on the regulatory mechanism in macrophages (Figure 1). Through the above studies, we attempt to provide a theoretical basis for cell-free therapy involving the macrophage regulation of inflammatory diseases such as periodontitis.

## 2. Results

### 2.1. Effects of Lipopolysaccharide (LPS) Pretreatment on Periodontal Ligament Stem Cells (PDLSCs)

Pretreatment of PDLSCs with 0.1, 1.0 and 10.0 μg/mL LPS for 24, 48, 72 and 96 h in the CCK8 assay showed no significant inhibition on PDLSC proliferation but promoted proliferation at 24 and 48 h (Figure 2A). The mRNA expression of *indoleamine 2, 3-dioxygenase* (*IDO*) in the 0.1 μg/mL LPS group were higher than those in the untreated (Control) and 1.0 μg/mL LPS groups. The mRNA expression of *IL-6* and *IDO* in the 1.0 μg/mL LPS group was higher than that in the Control group, and the mRNA expression of *IL-8* was significantly higher than that in the Control and 0.1 μg/mL LPS groups. The expression of *IL-10*, *IDO*, *tumor necrosis factor-stimulated gene 6 protein* (*TSG-6*), *IL-6* and *IL-8* mRNA in 10.0 μg/mL LPS-pretreated PDLSCs significantly increased compared with the other three groups, and *TGF-β* expression increased in the 10.0 μg/mL LPS group compared with the Control and 1.0 μg/mL LPS groups (Figure 2B). Then, we collected the conditioned medium (CM) derived from PDLSCs pretreated with 0, 0.1, 1.0 and 10.0 μg/mL LPS and used this to culture pre-polarized M1 macrophages. The results showed that compared with M1 macrophages, CM derived from PDLSCs pretreated with different concentrations of LPS inhibited the mRNA expression of the M1 marker *inducible nitric oxide synthase* (*iNOS*) (*p* < 0.05), especially in the 1.0 and 10.0 μg/mL LPS groups. Meanwhile, CM derived from PDLSCs pretreated with 10.0 μg/mL LPS significantly promoted high mRNA expression of the M2 marker *arginase-1* (*Arg-1*) compared with other groups (Figure 3A). Western blot showed that CM derived from PDLSCs pretreated with the different concentrations of LPS could all inhibit the protein expression of iNOS in M1 macrophages, and the 10.0 μg/mL LPS group significantly promoted the expression of Arg-1 (Figure 3B).

### 2.2. Construction of PDLSC–SupraGel Culture System

Figure 4A shows the chemical structure of the Biotin–DFYIGSRGD peptide that could self-assemble into SupraGel (Figure 4B). Untreated PDLSCs and PDLSCs pre-treated with 10.0 μg/mL LPS were encapsulated and cultured in SupraGel as three-dimensional (3D) cultures and in cell culture plates as two-dimensional (2D) culture controls. After vortexing, SupraGel could turn into a colloidal sol and mix with a suspension of PDLSCs to obtain a stable hydrogel with a cell density of 200,000 cells/mL after 20 min incubation at 37 °C (Figure 4C). Within 7days of culture, the cells extended in fusiform or polygon shapes in the 2D culture, while cells in SupraGel displayed no significant protrusion shapes (Figure 5A,B). Live/Dead imaging and semi-quantitative analysis indicated good viability of PDLSCs on days 1, 3, and 7 in both 2D and 3D culture conditions (Figure 3B,C). In addition, within 7 days, PDLSCs efficiently grew into small spheroids and gradually became larger with time (Figure 5B and Appendix A). However, there was no significant increase in the volume of PDLSCs spheroids on day 10, and they showed a tendency to gradually disperse and disassemble (Appendix A). The CCK8 assay revealed that PDLSCs showed an increased proliferation in culture plates within 7 days and significant inhibition on day 10 (Figure 5D). However, in 3D culture, PDLSCs with or without LPS pretreatment proliferated slowly, but did not show proliferation inhibition within 10 days of culture (Figure 5D).

### 2.3. Effects of Secretome on Macrophage Polarization and Migration

By addition of PBS and centrifugation, we successfully separated PDLSCs from SupraGel and collected the secretome derived from PDLSCs grown under different conditions: PDLSCs in cell culture plates (2D), LPS-pretreated PDLSCs in cell culture plates (2D/LPS), PDLSCs in SupraGel (3D), and LPS-pretreated PDLSCs in SupraGel (3D/LPS) (Figure 6A). The absence of cells during imaging by light microscopy and live/dead staining indicated complete cell removal from the supernatant (Figure 6B). Then, we detected the secretory protein concentration of the above four secretome on days 0, 3, 5, 7 and 10. At all time points, the protein content in the secretome from 3D culture was higher than those from 2D culture, and there was no correlation with LPS pretreatment (Figure 6C). Each of the different secretome was then used to culture M1 macrophages. The mRNA expression of *iNOS*, *IL-1β* and *IL-6* significantly decreased in M1 macrophages cultured with secretome derived from the 2D, 2D/LPS, 3D and 3D/LPS groups compared with M1 macrophages. The mRNA expression of *tumor necrosis factor-α* (*TNF-α*) in M1 macrophages cultured with 2D-, 3D- and 3D/LPS-group-derived secretome was significantly decreased compared with its expression in M1 macrophages. Except for *IL-6*, the 3D/LPS-group-derived secretome did not show more effective suppression of inflammatory factors in M1 macrophages compared with secretome derived from the 2D, 2D/LPS and 3D groups (Figure 7A). The mRNA expression of *Arg-1* was significantly higher in 2D/LPS-group-derived-secretome-cultured M1 macrophages than in M1 macrophages and 2D-group-derived-secretome-cultured M1 macrophages, and in 3D/LPS-group-derived-secretome-cultured M1 macrophages than in M1 macrophages and both 2D- and 3D-group-derived-secretome-cultured M1 macrophages (Figure 7B). The mRNA expression of *IL-10* was significantly higher in M1 macrophages cultured with the 2D/LPS-group-derived secretome than in M1 macrophages, and in 3D/LPS-group-derived-secretome-cultured M1 macrophages than in M1 macrophages cultured with secretome derived from other groups (Figure 7B). Western blotting showed that secretome derived from the 2D, 2D/LPS, 3D, 3D/LPS groups inhibited iNOS and CD86 expression in M1 macrophages; the 3D/LPS treatment group had the strongest inhibitory ability on iNOS, while the 2D/LPS treatment group showed the strongest inhibition of CD86. In addition, 2D/LPS- and 3D/LPS-group-derived secretome promoted Arg-1 and CD206 expression in M1 macrophages compared with other groups (Figure 7C and Appendix A). Furthermore, the results of the scratch assay showed that the 3D/LPS-group-derived secretome significantly promoted the migration of macrophages at 24 h compared with the FBS-free medium, 2D and 3D groups. At 48 h, 2D/LPS- and 3D/LPS-group-derived secretome significantly promoted the migration of macrophages (Figure 7D,E).

We performed an immune cytokines assay (48 cytokines) on 2D-, 2D/LPS-, 3D- and 3D/LPS-group-derived secretome to study whether the different treatments translated into an altered paracrine content in the corresponding secretome. The expression levels of different cytokines are summarized in Figure 8 and Figure 9 and Appendix A. Of the 48 cytokines examined, 32 significantly changed after LPS and/or 3D culture treatment. Firstly, the secretion of IL-6, IL-8, IL-4, IL-13, IL-17, M-CSF, IFN-γ and GM-CSF involved in macrophage polarization significantly increased in PDLSCs after LPS and/or 3D culture treatment (Figure 8A). IL-4 increased in the 3D/LPS-group-derived secretome, IL-13 increased in 3D- and 3D/LPS-group-derived secretome, while IL-6 and IL-8 increased in 2D/LPS-, 3D- and 3D/LPS-group-derived secretome, especially in the 2D/LPS and 3D/LPS groups. Another cytokine that increased in PDLSCs after 3D culture is IL-17A. In addition, M-CSF increased in the 2D/LPS-group-derived secretome and GM-CSF increased in the 3D- and 3D/LPS-group-derived secretome. Meanwhile, significant changes in the expression of G-CSF, IL-3, SCF in PDLSCs were observed after pretreatment (Figure 8A). The production of G-CSF and IL-3 increased and SCF decreased in the 3D-cultured PDLSC-derived secretome. The secretion of multiple chemokines significantly changed in PDLSC-derived secretome after LPS pretreatment and/or 3D culture (Figure 8B). MCP-1 increased in 2D/LPS- and 3D/LPS-group-derived secretome but decreased in the 3D-group-derived secretome, and MCP-3 and MIG increased in the 3D/LPS-group-derived secretome. MIP-1α and Eotaxin increased in the 3D-group-derived secretome. 3D culture promoted the secretion of IL-16 and IP-10 in PDLSCs, while LPS pretreatment promoted the expression of RANTES and GRO-α in PDLSCs. MIP-1β and CTACK secretion by PDLSCs increased after LPS and/or 3D culture treatment. The secretion of SDF-1α by PDLSCs decreased after 3D culture. In addition, we noticed that after 3D culture, the expression of VEGF, HGF, PDGF and SCGF-β increased in PDLSC-derived secretome (Figure 9A). In addition to the aforementioned cytokines, IL-9 increased in 2D/LPS-, 3D- and 3D/LPS-group-derived secretome and IL-2Rα increased in 3D- and 3D/LPS-group-derived secretome. The inflammatory cytokines IL-1α and IFN-α2 increased in 3D- and 3D/LPS-group-derived secretome, and TNF-β increased in 2D/LPS- and 3D/LPS-group-derived secretome (Figure 9B).

IL-1β, IL-Ra, IL-2, IL-5, IL-7, IL-10, IL-12(P40), IL-12(P70), IL-15, IL-18, TNF-α, MIF, β-NGF, basic-FGF, TRAIL and LIF showed no significant changes in the four group-derived secretome (Appendix A).

## 3. Discussion

As a commonly used inflammatory inducer, LPS has been studied in MSC pretreatment. As mentioned above, LPS-pretreated MSC-derived exosomes could induce the polarization of M2 macrophages and secrete anti-inflammatory factors, thus playing a role in the treatment of inflammatory diseases. However, some studies have shown that LPS could also promote the pro-inflammatory phenotype of MSCs and reverse their therapeutic immunosuppressive effects [33]. LPS-pretreated human thymic MSC-derived exosomes promoted the polarization of M1 macrophages, the production of IL-6 and TNF-α and the differentiation of Th17 cells [34]. This immunomodulatory difference of LPS-pretreated MSCs may be caused by cell types (mouse versus human), tissue origin, concentration of LPS, in vivo versus in vitro studies and pretreatment time [8]. A study indicated that LPS pretreatment for different times could change the pro-inflammatory and anti-inflammatory phenotype of MSCs, which then played different immunomodulatory and therapeutic roles [35]. In this study, we set a concentration gradient containing the concentration of LPS and a fixed time point commonly used in research to preliminarily explore the effects of LPS concentration on PDLSCs. We found that the mRNA expression of *IL-10*, *IDO*, *TGF-β*, *TSG-6*, *IL-6* and *IL-8* in PDLSCs pretreated with 10.0 μg/mL LPS was significantly increased. Research has indicated that the expression of IL-10, IDO, TGF-β, TSG-6, IL-6 and IL-8 in MSCs could promote the polarization of M2 macrophages [36,37,38,39,40]. Thus, we speculated that of the PDLSC-derived secretome following 10.0 μg/mL LPS pretreatment could have greater potential to induce M2 macrophage polarization. M1 macrophages were cultured in CM derived from PDLSCs pretreated with different concentrations of LPS. The results showed that the CM derived from 10.0 μg/mL LPS-pretreated PDLSCs had the most obvious ability to promote the polarization of macrophages from an M1 to M2 subtype, which verified our hypothesis. Based on the results, 10.0 μg/mL LPS was chosen as the inflammatory stimulus for the subsequent experiments.

MSCs are typically grown as a monolayer on 2D plastic culture plates for ex vivo expansion. However, removing them from their endogenous 3D niche as well as enzymatic passaging result in loss of multipotency, replicative senescence, decreased cytokine production and expression of surface markers (e.g., CD105, CD90, CD73) that are associated with the MSCs’ undifferentiated phenotype [13,28,41]. MSCs in vivo are exposed to mechanophysical signals in all three dimensions of the surrounding niche; therefore, spheroids or a hydrogel culture surrounding cells may provide stronger inductive signals to modulate the properties and function of MSCs compared with 2D culture environments [28]. Compared with cell culture plates, secretion of most cytokines increased when either early or late passage MSCs were cultured on soft hydrogels [41]. Studies on senescence-related changes showed that compared with 2D culture, senescence-related β-galactosidase activity significantly decreased, and telomerase activity and telomere length increased in 3D culture [42]. In addition, MSC spheroids are common 3D culture methods and efficiently exert immunomodulatory effects [43,44]. However, there is still some debate on whether the MSC spheroids would affect cells proliferation and viability when they reach critical sizes followed by non-homogenous nutrient and oxygen supply [45]. Similarly, the proliferation and viability of 3D culture of MSCs on scaffolds and hydrogels are also complex, which might be related to their composition, preparation methods, and the source of MSCs [28]. Despite the low proliferative activity of PDLSCs under 3D culture conditions, no significant cell necrosis was observed, which might play an important potential role in preserving their stemness, pluripotency and viability, preventing replicative senescence and affecting secretion. In addition, the cells showed good viability whether they were cultured in hydrogel or in cell culture plates over 7 days, but 2D cultured cells showed proliferation inhibition on day 10. Therefore, in order to avoid the influence of cell viability on the experiment, day 7 was selected as the final time point for culture. Summing up the above, we successfully constructed a PDLSC–SupraGel culture system.

In the studies of secretome collection after 3D cell culture using hydrogels, most of them collected the culture supernatant, i.e., CM [12,46]. However, the secretome is not only present in the CM, but also inevitably exists in hydrogel and it would be a loss if this part of the secretome was abandoned. However, since the cells are inside the gel, complete separation of the cells from the gels is key to collecting the whole secretome. In this study, PDLSCs were successfully removed from SupraGel by dissolving SupraGel with PBS followed by centrifugation, enabling the collection of all the secretome. In order to fully remove the cells, we adjusted the centrifugal speed and time, and the final determined parameters were different from the previous study [29], which may be caused by the differences in cell types and the purpose of the experiment. The previous study used tumor cell lines and intestinal stem cells, and the purpose of that study was to collect cells spheroids in the hydrogel. Therefore, in order to ensure the integrity of the spheroids, the speed used was relatively low. In this study, to ensure the complete removal of cells, the speed was increased to 3000 rpm; however, at this speed, the cells spheroids and cells were damaged. Therefore, PDLSCs obtained by centrifugation after 3D culture could not be used to detect stemness, pluripotency and senescence. Further experiments are needed to improve the parameters of centrifugation.

At all time points, the protein content in the secretome of 3D cultures was higher than those of 2D cultures, which was consistent with the idea that 3D culture may influence the secretion of MSCs. In this part, it is worth noting that, although the number of initial seeded cells was consistent, due to the different cell proliferation in 2D and 3D culture, the different number of cells at each time point would affect the amount of secreted protein. We tried to calculate the number of cells after 3D and 2D culture in an attempt to assess the amount of secreted protein by eliminating the effect of differences in cell numbers, but the cells after 3D culture showed clusters and could not be counted accurately. According to the results of the CCK8 assay in Section 2.2, the cell proliferation of 2D cultures was significantly higher than that of 3D cultures over 7 days. If calculated based on protein quantity/cell number, the amount of protein secreted by one PDLSC in 3D culture would be much higher than that in 2D culture. Therefore, it was plausible to conclude that the secretory activity of PDLSCs was significantly promoted by 3D culture compared to 2D culture.

The results of different secretome on macrophage polarization indicated that 3D-cultured PDLSC-derived secretome enhanced the inhibitory effect on M1 macrophage polarization and that LPS pretreatment of PDLSC-derived secretome had a strong ability to promote the polarization of M1 to M2 macrophages. A study indicated that MSC spheroids increased the secretion of prostaglandin E2, IDO, TGF-β1 and IL-6 and the immunosuppressive effect on the functional activity of macrophages, which was further enhanced by IFN-γ and TNF-α and dependent on fetal bovine serum (FBS) in the cell culture medium [27]. To collect pure secretome, we cultured cells in FBS-free medium, possibly resulting in insufficient secretion of macrophage polarization regulators without LPS stimulation. Therefore, the polarization of M1 to M2 macrophages was not significant in 2D and 3D groups. The migration results indicated that secretome derived from LPS-pretreated PDLSCs had an enhanced chemotactic effect on macrophages, while the effect of 3D culture was not obvious. To sum up, the PDLSC-derived secretome after LPS pretreatment and/or 3D culture had regulatory effects on macrophage. However, the factors that regulate macrophage polarization and migration are still unknown, and the composition changes of the secretome in different treatment groups need to be detected to explore the possible mechanisms.

By detecting the immune cytokines in the secretome, we found that a variety of cytokines involved in macrophage polarization are significantly increased in PDLSCs after LPS and/or 3D culture treatment, and that the two different treatment methods induced the production of different cytokines that regulate macrophage polarization. IL-4 and IL-13 play important roles in the polarization of M2 macrophages by activating the JAK3/STAT6 signaling pathway [6]. IL-6 and IL-8 could enhance macrophage differentiation into the M2 subtype by activating the STAT3 signaling pathway [40,47,48,49]. Moreover, IL-6 promoted IL-4 and IL-13 binding to the IL-4 receptor (IL-4R) by upregulating IL-4R expression, and it subsequently activated STAT6/PPAR γ signaling pathway, ultimately inducing M2-type polarization [49,50]. In our previous results, 2D/LPS- and 3D/LPS-group-derived secretome promoted macrophage polarization to an M2 type. We speculated that, on the one hand, high expression of IL-4, IL-13, IL-6 and IL-8 promoted macrophage polarization to an M2 type, while on the other hand, high expression of IL-6 increased the binding of IL-4 and IL-13 to IL-4R, which further promoted the polarization of M2 macrophages. However, IFN-γ and GM-CSF increased in 3D- and 3D/LPS-group-derived secretome, and they could activate M1 macrophages [51,52]. IL-17A had contradictory effects on macrophage polarization. It could mediate macrophage recruitment and directly or indirectly induce M2 macrophage polarization in inflammatory diseases [53,54], but it also could enhance IFN-γ-induced M1 macrophage polarization while suppressing IL-4-mediated M2 transformation [55]. GM-CSF was associated with M1 macrophage polarization while M-CSF was linked with M2 macrophage polarization, but when compared with prototypic polarizing stimulation (e.g., IFN-γ, LPS, IL-4, IL-10, etc.), neither of these two factors were potent stimulators for definitive polarization markers [56]. Meanwhile, GM-CSF, G-CSF, IL-3, IL-6 and SCF were all associated with the production of monocytes and macrophages derived from hematopoietic stem cells [57]. Thus, 3D- and 3D/LPS-group-derived secretome have the potential to induce more monocyte/macrophage production in vivo; however, this needs further validation. The secretion of chemokines, which are involved in promoting the migration and trafficking of immune cells, including monocytes and macrophages [58,59,60,61], were significantly changed after LPS and/or 3D culture. Compared with LPS pretreatment, 3D culture caused a reduction in some chemokines, which might induce an insignificant chemotactic effect on macrophages. In addition, the high expression of growth factors are involved in the survival and self-renewal of hematopoietic stem cells, angiogenesis, osteogenesis and immunomodulation [62,63,64,65,66], which indicates that the secretome derived from 3D-cultured PDLSCs had enhanced tissue regeneration potential. In summation, we hold the opinion that if the secretome from 3D-cultured LPS-pretreated PDLSCs is used on local sites of inflammation in vivo, the chemokines might induce the aggregation of monocytes/macrophages closely associated with inflammatory sites, and the cytokines involved in macrophage polarization might cause these cells to polarize to M2 macrophages, thereby improving the inflammatory microenvironment. Subsequently, growth factors would promote the regeneration of inflammatory damaged tissues. In addition, the secretion of IL-9, IL-2Rα, IL-1α, TNF-β and IFN-α2 increased in PDLSC-derived secretome after 3D culture and/or LPS treatment. IL-9 is a pleiotropic cytokine that is involved in both protective immunity and immunopathology [67]. IL-2R signaling is indispensable for Treg cells lineage stability and suppressor function in inflammatory environments [68]. They may act in conjunction with other factors in the secretome. In the previous section, we thought that the slowing proliferation of PDLSCs in SupraGel was beneficial to maintain their stemness and pluripotency. We found that the expression trend of two key factors, HGF and SCF, in maintaining the stemness of MSCs was different after 3D culture [69]. Therefore, the maintenance of stemness and pluripotency of PDLSCs by 3D culture needs to be verified by collecting isolated cells from the SupraGel. Due to the high speed of centrifugation, the PDLSCs were damaged after centrifugation, so it is necessary to further improve the centrifugation method.

To sum up, in general, PDLSCs secreted more macrophage-regulating cytokines after LPS pretreatment and/or 3D culture. We have verified the effects of LPS- and/or 3D-culture-treated-PDLSC-derived secretome on macrophage polarization and migration, but the comparative effects of the specific components (one or more soluble factors or EVs) and the overall application of the secretome remains to be verified, which has great significance to guide its clinical use in the future.

In conclusion, we found that PDLSCs and 10.0 μg/mL LPS-pretreated PDLSCs had good survival vitality in SupraGel and that the gel could be used as a tool to collect the secretome. The expressions of cytokines involved in the polarization, differentiation and chemotaxis of macrophages and pro-regenerative growth factors in the PDLSC-derived secretome were increased after LPS pretreatment and/or 3D culture, which promoted M2 macrophage polarization and macrophage migration in vitro. The above results suggest that the secretome derived from 3D-cultured LPS-pretreated PDLSCs has the potential to regulate the transformation of an inflammatory microenvironment to a regenerative microenvironment by regulating macrophage migration and polarization. Importantly, further in vivo studies should be performed to verify its effect on inflammatory diseases such as periodontitis.

## 4. Materials and Methods

### 4.1. Proliferation Assay of Periodontal Ligament Stem Cells (PDLSCs)

Human PDLSCs used in this study were purchased from Procell Life Science&Technology Co., Ltd. (Wuhan, China), and grown in α-MEM medium (Gibco, Thermo Fisher, Waltham, MA, USA) supplemented with 10% fetal bovine serum (FBS) (Gibco, Thermo Fisher, Waltham, MA, USA) and 1% Penicillin–Streptomycin (P/S) (Gibco, Thermo Fisher, Waltham, MA, USA). Cells were used before passage 5. PDLSCs were harvested by trypsinization and inoculated into a 96-well plate at a density of 2000 cells per well. After cell adherence, 100 μL medium with different concentrations of lipopolysaccharide (LPS) (0, 0.1, 1.0, 10.0 μg/mL) was added to each well, then after 24, 48, 72, 96 h, 10 μL CCK8 reagent (Biosharp, Hefei, China) was added and incubated at 37 °C for 2 h. The absorbance was measured using a microplate reader (BioTek, Winooski, VT, USA) at 450 nm. The CCK8 reagent without cells was the blank.

### 4.2. Macrophage Polarization

Mouse mononuclear macrophage leukemia cells (RAW264.7 cells) were purchased from Procell Life Science&Technology Co., Ltd. (Wuhan, China), and inoculated into a 6-well plate at a density of 100,000 cells per well, and then polarized to M1 macrophages by pretreatment with 20 ng/mL interferon (IFN)-γ (R&D Systems, Minneapolis, MN, USA) for 24 h.

### 4.3. LPS Pretreatment of PDLSCs and Conditioned Medium (CM) Culture of M1 Macrophages

PDLSCs were harvested by trypsinization and inoculated into a 6-well plate at a density of 200,000 cells per well. After cell adherence, 2 mL medium with different concentrations of LPS (0, 0.1, 1.0, 10.0 μg/mL) was added to each well and cultured for 48 h. Conditioned medium (CM) derived from PDLSCs pretreated with different concentrations of LPS (0, 0.1, 1.0, 10.0 μg/mL) was collected to culture pre-polarized M1 macrophages.

### 4.4. Construction of a PDLSC–SupraGel Culture System

When PDLSCs and LPS-pretreated PDLSCs reached 80–90% confluency, the medium was removed, and the cells were washed twice with phosphate-buffered saline (PBS) and replaced with fresh culture medium containing LPS (10.0 μg/mL). The culture medium without LPS was used as a control. The culture was followed for another 48 h. Then PDLSCs were harvested by trypsinization, washed by PBS and a single-cell suspension (600,000 cells/mL) obtained. A 100 μL aliquot of SupraGel was vortexed, and then mixed with 50 μL of the cell suspension to obtain a final cell density of 200,000 cells/mL. The cell–gel solution (150 µL) was transferred into a 48-well plate. The plate was then incubated for 20 min at 37 °C to stabilize the gels. Once gels were formed, 200 µL warm α-MEM medium without FBS was added on top of the cell–gel constructs. As a control, a 2D culture was prepared by seeding a 48-well plate with 200 µL of the cell suspension (same total cell number as the 3D culture). Four groups were divided and named as follows: PDLSCs in cell culture plate was the 2D group, LPS-pretreated PDLSCs grown in a cell culture plate was the 2D/LPS group, PDLSCs in SupraGel was the 3D group and LPS-pretreated PDLSCs in SupraGel was the 3D/LPS group. Each well was replaced with 200 µL fresh medium every other day. Cell culture supernatants (i.e., CM) was collected at days 0, 3, 5, 7 and 10 and secreted protein concentration was measured by a BCA protein kit (Beyotime, Shanghai, China).

### 4.5. PDLSC Viability in SupraGel

Live/Dead staining was performed using a Calcein/PI cell activity and cytotoxicity assay kit (Beyotime, Shanghai, China). In brief, cell culture medium was gently removed and 150 μL Calcein AM/PI working solution was prepared and added to the samples on days 1, 3 and 7. After 30 min incubation away from light, all samples were imaged by laser scanning confocal microscope (LSM800, ZEISS, Jena, Germany). The Live/Dead merge images of at least three repeated wells in each group were selected and every image was divided into three channels. The mean gray values of green and red fluorescence were measured separately by Image J software (NIH, Bethesda, MD, USA) and compared with the total gray value to obtain the percentage of live and dead cells, respectively.

At days 1, 3, 7 and 10, 20 μL CCK8 reagent was added to each well and incubated at 37 °C for 2 h, and then 100 µL of supernatant working solutions were transferred to another 96-well plate. The absorbance was measured using a microplate reader (BioTek, Winooski, VT, USA) at 450 nm.

### 4.6. Collection of Secretome

On day 7, the PDLSC–SupraGel were transferred from the cell culture plate to a 1.5-mL eppendorf tube. An aliquot of 150 μL PBS was added to the eppendorf tube and resuspended, and then the mixture was centrifuged at 3000 rpm for 3 min. The supernatant was gently aspirated and mixed with the CM collected on days 3, 5 or 7 and then stored at −80 °C. The secretome collected above was used for the experiments of macrophage polarization and migration and for the detection of immune cytokines.

### 4.7. Secretome Culture of M1 Macrophages

RAW264.7 cells were inoculated into a 6-well plate at a density of 500,000 cells per well, and then polarized to M1 macrophages. Secretome derived from 2D, 2D/LPS, 3D and 3D/LPS groups were collected and cultured with M1 macrophages for 24 h.

### 4.8. Wound Healing

RAW264.7 cells were seeded into 6-well plates at a density of 500,000 cells/mL. Once a confluent monolayer was observed, a wound was made by dragging a plastic pipette tip across the cell surface. The remaining cells were washed three times with PBS to remove cell debris and incubated at 37 °C with FBS-free medium and 2D-, 2D/LPS-, 3D-, 3D/LPS-group-derived secretome. The wound closure was monitored and photographed at 24 and 48 h. Quantitative analysis of wound closure was measured with Image J software (NIH, Bethesda, MD, USA).

### 4.9. Quantitative Reverse Transcription Polymerase Chain Reaction (qRT-PCR)

Total RNA was extracted from the cells using the cell total RNA isolation kit (Foregene, Chengdu, China) according to the manufacturer’s instructions. The RNA samples were stored in the refrigerator at −80 °C. Complementary DNA was synthesized using the PrimeScript RT reagent Kit (Takara, Tokyo, Japan) following the manufacturer’s protocol. Then qRT-PCR was performed using a Hieff qPCR SYBR Green Master Mix kit (Yeasen, Shanghai, China) and a LightCycler 96 (Roche Applied Science, Indianapolis, IN, USA). The sequences of primers are listed in Table 1. The expression of target genes was normalized to GAPDH and expressed as fold-change using the 2^−∆∆CT^ method.

### 4.10. Western Blot

The total protein of cells was extracted using radioimmunoprecipitation assay lysis buffer (Beyotime, Shanghai, China). Protein concentrations were measured using the BCA protein kit (Beyotime, Shanghai, China). Proteins were separated by electrophoresis using 5–15% SDS-PAGE gels and transferred to a poly (vinylidene fluoride) (PVDF) membrane (Millipore, St. Louis, MI, USA). The PVDF membrane was blocked by soaking in 5% skimmed milk and then incubated at 4 °C overnight with primary antibodies as follows: β-actin (bs-0061R, Bioss, Beijing, China, 1:10,000), iNOS (18985-1-AP, Proteintech, Wuhan, China, 1:600), Arg-1 (16001-1-AP, Proteintech, Wuhan, China, 1:20,000), CD86 (13395-1-AP, Proteintech, Wuhan, China, 1:600), CD206 (sc-70585, Santa Cruz, Santa Cruz, CA, US, 1:200). The PVDF membranes were washed three times with Tris-buffered saline containing Tween and incubated with horseradish peroxidase-conjugated secondary antibody (1:5000) at room temperature for 1 h. The enhanced chemiluminescence solution was mixed with the stable peroxidase solution (ratio of 1:1), which was then dropped onto the PVDF membrane. Finally, imaging was performed in the imaging system, and the results were observed and recorded.

### 4.11. Multiplex Cytokine Assay of the Secretome

The secretome was centrifugated at 1000× *g* for 15 min at 4 °C and then sent to Applied Protein Technology (Shanghai, China) for multiplex cytokine analysis. The multiplex immunoassays are based on fluorescently dyed magnetic beads for selected cell culture supernatant biomarkers from a Bio-Plex Pro™ Human Cytokine Screening Panel (48-Plex #12007283). Briefly, the antibody-coupled magnetic capture beads were first incubated with antigen standards or samples for a specific time. The plate was then washed to remove unbound materials and incubated with biotinylated detection antibodies. After washing away unbound antibodies, the beads were incubated with a reporter streptavidin–phycoerythrin conjugate (SA–PE). After removal of excess SA–PE, the beads were passed through the array reader, thereby measuring the fluorescence of the bound SA–PE. A total of 48 cytokines were quantified: Cutaneous T-cell attracting chemokine (CTACK)/chemokine (C-C motif) ligand (CCL) 27, Eosinophil chemotactic protein (Eotaxin)/CCL11, basic fibroblast growth factor (Basic-FGF), granulocyte colony-stimulating factor (G-CSF), granulocyte-macrophage colony-stimulating factor (GM-CSF), growth related oncogene-alpha (GRO-α)/chemokine (C-X-C motif) ligand (CXCL) 1, hepatocyte growth factor (HGF), IFN-α2, IFN-γ, interleukin (IL)-1α, IL-1β, IL-1 receptor agonist (IL-1Ra), IL-2, IL-2Ra, IL-3, IL-4, IL-5, IL-6, IL-7, IL-8/CXCL8, IL-9, IL-10, IL-12 p40 subunit (IL-12 (p40)), IL-12(p70), IL-13, IL-15, IL-16, IL-17, IL-18, IFN-inducible protein 10 (IP-10)/CXCL10, leukemia inhibitory factor (LIF), monocyte chemoattractant protein (MCP)-1/CCL2, MCP-3/CCL7, macrophage colony-stimulating factor (M-CSF), macrophage migration inhibitory factor (MIF), monokine induced by IFN-γ (MIG)/CXCL9, macrophage inflammatory protein (MIP)-1α/CCL3, MIP-1β/CCL4, β-nerve growth factor (β-NGF), platelet-derived growth factor bb (PDGF-BB), regulated on activation, normal T cell expressed and secreted (RANTES)/CCL5, stem cell factor (SCF), stem cell growth factor-β (SCGF-β), stromal cell-derived factor-1α (SDF-1α)/ CXCL12, tumor necrosis factor (TNF)-α, TNF-β, TNF-related apoptosis-inducing ligand (TRAIL) and vascular endothelial growth factor (VEGF).

### 4.12. Statistical Analysis

Statistical analysis was performed with SPSS 20.0 (IBM SPSS, Chicago, IL, USA). All data are expressed as mean ± standard deviation (SD) of at least three independent experiments. The figures were generated by GraphPad Prism 8.0.2 (GraphPad Software, Boston, MA, USA). The Kolmogorov–Smirnov test was used to assess whether continuous data were normally distributed. For the data of continuous numerical variables that conform to a normal distribution and have homogeneity of variance, we carried out variance analysis. For the data that did not exhibit a normal distribution, a nonparametric test was used. In all cases, *p* values < 0.05 derived from a 2-tailed test were considered statistically significant.

## Figures and Tables

**Figure 1 ijms-24-06981-f001:**
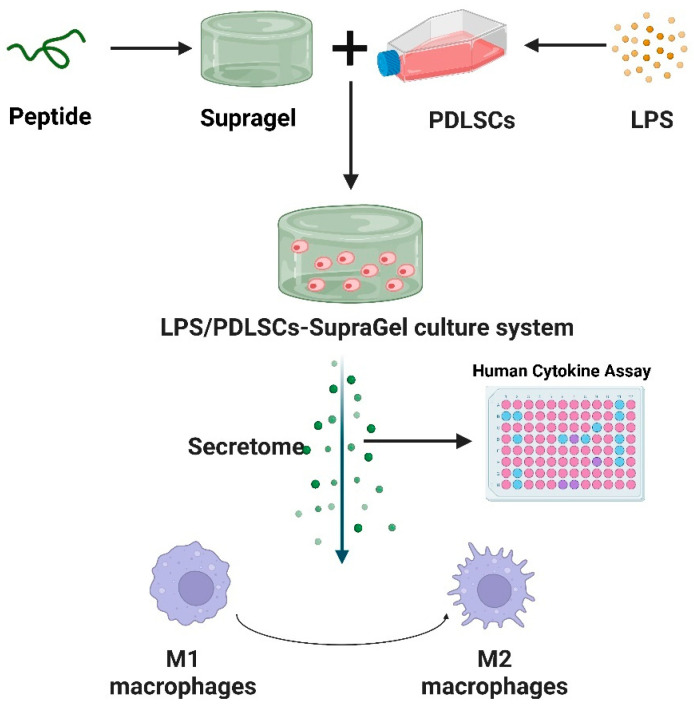
Schematic diagram of 3D-cultured LPS-pretreated PDLSC-derived secretome on macrophage polarization; created with BioRender.com (accessed on 27 March 2023).

**Figure 2 ijms-24-06981-f002:**
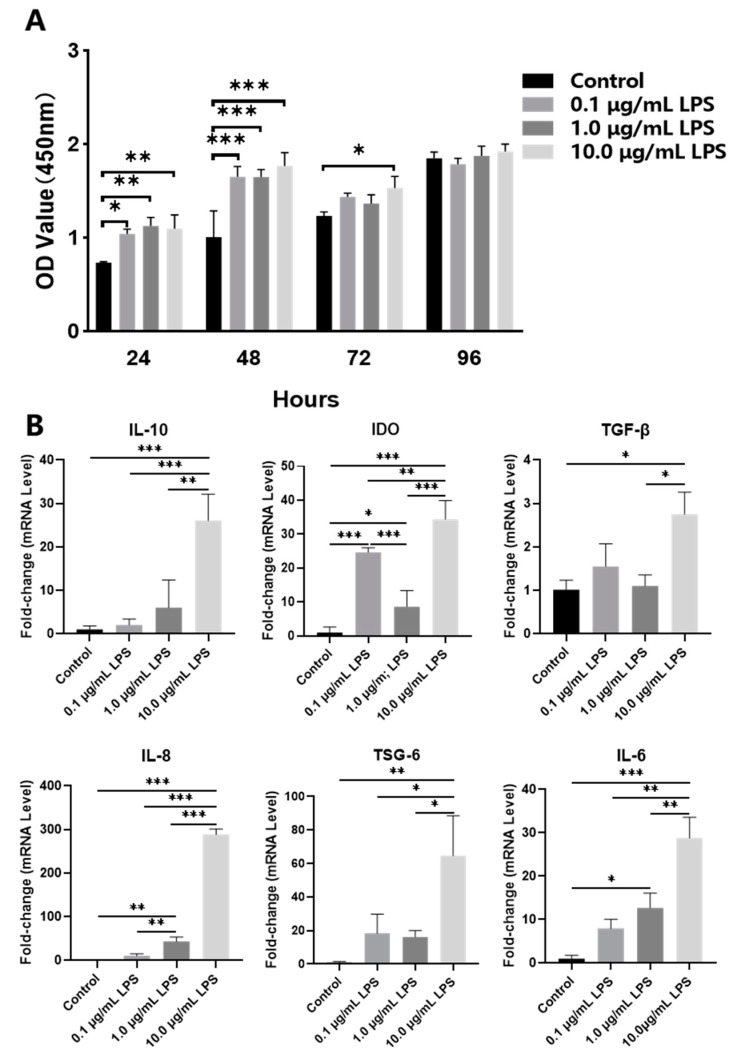
Effect of LPS pretreatment on PDLSCs. (**A**) CCK8: PDLSC proliferation after culturing with 0 (Control), 0.1, 1.0 and 10.0 μg/mL LPS for 24, 48, 72 and 96 h. (**B**) qRT-PCR: *IL-10*, *IDO*, *TSG-6*, *IL-6*, *IL-8* and *TGF-β* mRNA expression in PDLSCs after culturing with 0 (Control), 0.1, 1.0 and 10.0 μg/mL LPS for 48 h. * *p* < 0.05; ** *p* < 0.01; *** *p* < 0.001.

**Figure 3 ijms-24-06981-f003:**
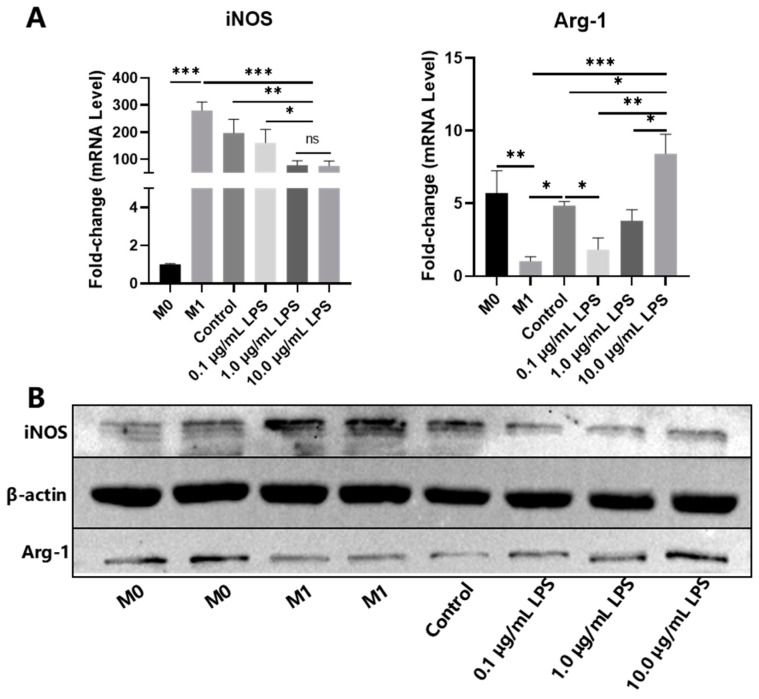
Effect of CM derived from LPS-pretreated PDLSCs on macrophage polarization. (**A**) qRT-PCR: *iNOS* and *Arg-1* mRNA expression in M1 macrophages after culturing with CM derived from 0 (Control), 0.1, 1.0 and 10.0 μg/mL LPS-pretreated PDLSCs. (**B**) Western blot: iNOS and Arg-1 protein expression in M1 macrophages after culturing with CM derived from 0 (Control), 0.1, 1.0 and 10.0 μg/mL LPS-pretreated PDLSCs. * *p* < 0.05; ** *p* < 0.01; *** *p* < 0.001.

**Figure 4 ijms-24-06981-f004:**
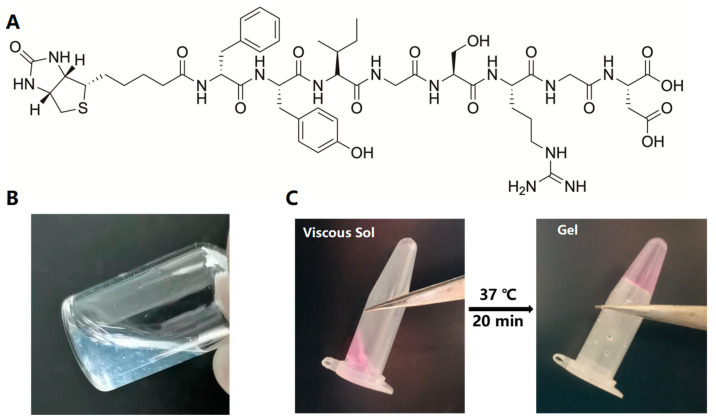
Construction of the PDLSC–SupraGel culture system. (**A**) Chemical structure of the Biotin–DFYIGSRGD peptide. (**B**) Peptide self-assembly to form SupraGel. (**C**) Sol of the SupraGel with PDLSC suspension becoming a stable hydrogel after 20 min incubation at 37 °C.

**Figure 5 ijms-24-06981-f005:**
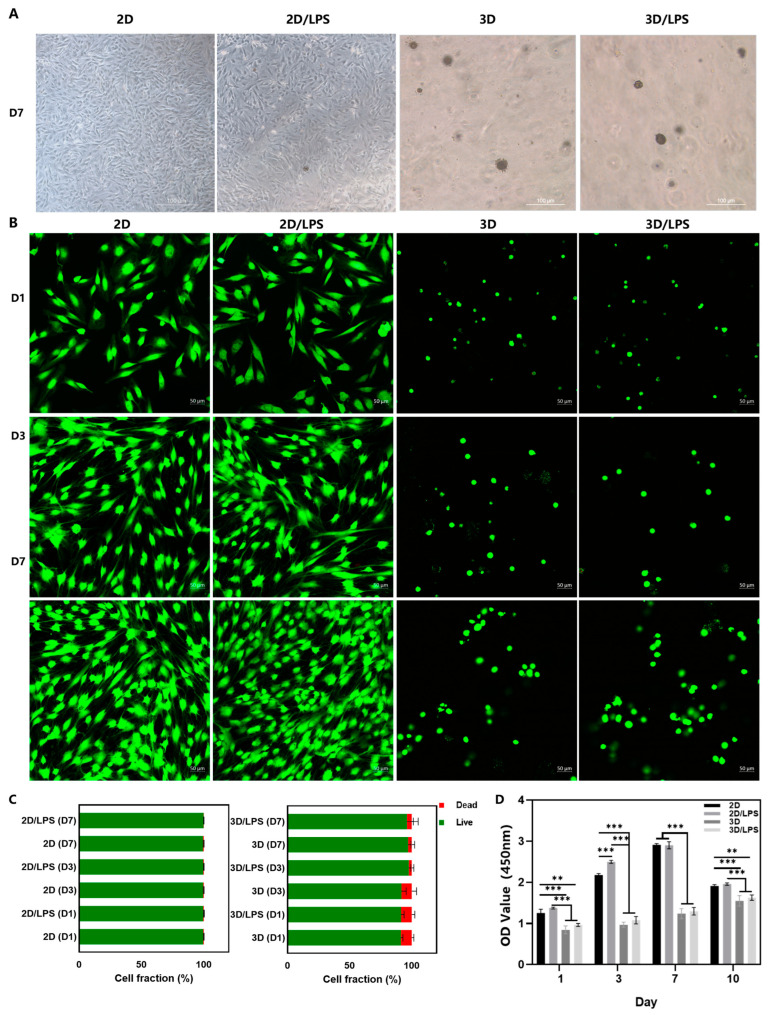
Growth, viability and proliferation of PDLSCs in SupraGel. (**A**) Morphology of PDLSCs and LPS-pretreated PDLSCs under light microscope; scale bar—100 μm. (**B**) Live/Dead image under laser scanning confocal microscope; scale bar—50 μm, green—live cells, red—dead cells. (**C**) Semi-quantitative analysis of the Live/Dead assay. (**D**) CCK8 proliferation assay of PDLSCs and LPS-pretreated PDLSCs after culturing in culture plates and SupraGel on days 1, 3, 7 and 10. 2D—PDLSCs in cell culture plate; 2D/LPS—LPS-pretreated PDLSCs in cell culture plate; 3D—PDLSCs in SupraGel; 3D/LPS—LPS-pretreated PDLSCs in SupraGel. D1—Day 1; D3—Day 3; D7—Day 7. ** *p* < 0.01; *** *p* < 0.001.

**Figure 6 ijms-24-06981-f006:**
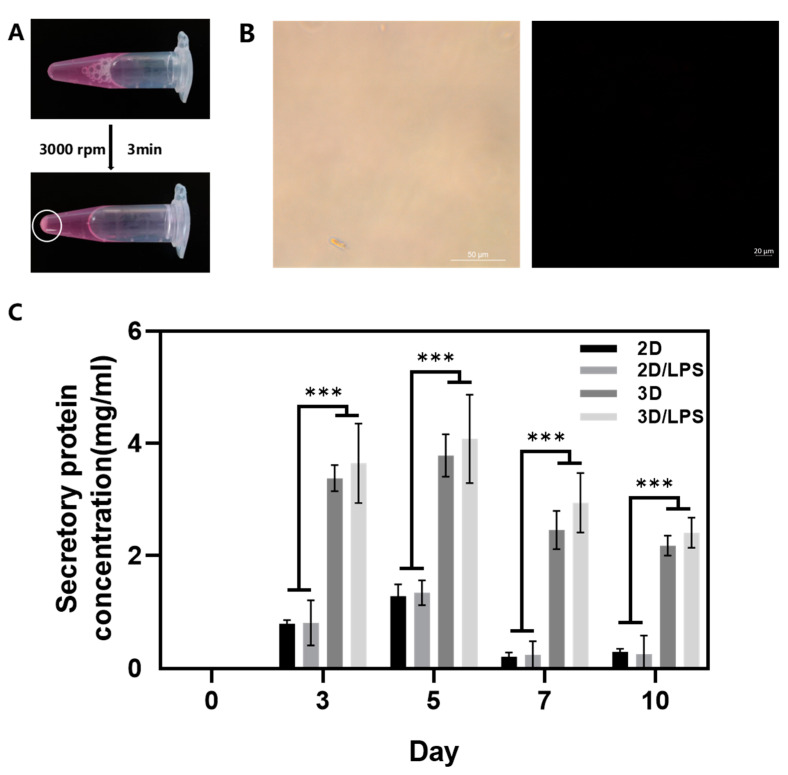
Collection and protein concentration measurement of secretome. (**A**) Separation of PDLSCs and SupraGel by adding PBS and centrifugation; white circle shows cell deposits. (**B**) Images of the supernatant after centrifugation. Left: light microscope image; scale bar—50 μm; Right: Live/Dead image; scale bar—20 μm. (**C**) Secretory protein concentration of 2D-, 2D/LPS-, 3D- and 3D/LPS-group-derived secretome on days 0, 3, 5, 7 and 10. *** *p* < 0.001.

**Figure 7 ijms-24-06981-f007:**
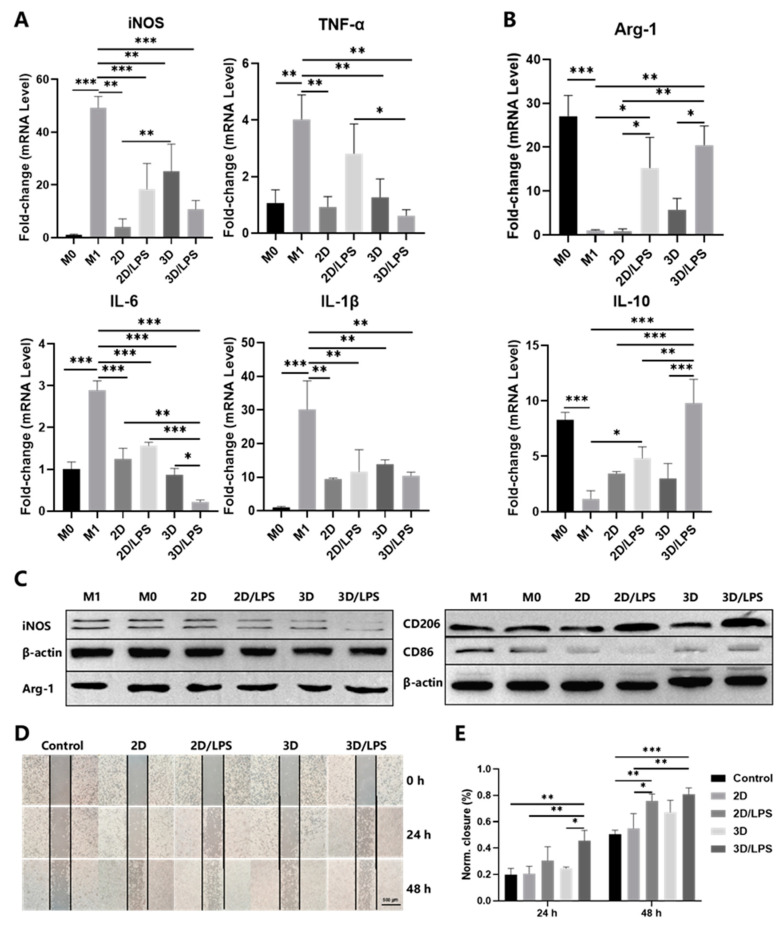
Effects of the secretome on macrophage polarization and migration. (**A**) qRT-PCR: *iNOS*, *TNF-α*, *IL-6* and *IL-1β* mRNA expression in M0, M1 and 2D-, 2D/LPS-, 3D-, 3D/LPS-group-derived-secretome-cultured M1 macrophages. (**B**) qRT-PCR: *Arg-1* and *IL-10* mRNA expression in M0, M1 and 2D-, 2D/LPS-, 3D- and 3D/LPS-group-derived-secretome-cultured M1 macrophages. (**C**) Western blot: iNOS, Arg-1, CD86 and CD206 expression in M0, M1 and 2D-, 2D/LPS-, 3D-, 3D/LPS-group-derived-secretome-cultured M1 macrophages. (**D**) Wound healing of macrophages cultured with FBS-free medium (Control) and 2D-, 2D/LPS-, 3D- and 3D/LPS-group-derived secretome for 0, 24 and 48 h under a microscope; scale bar—500 μm. (**E**) Wound closure of macrophages cultured with FBS-free medium (Control) and 2D-, 2D/LPS-, 3D-, 3D/LPS-group-derived secretome for 0, 24 and 48 h. * *p* < 0.05; ** *p* < 0.01; *** *p* < 0.001.

**Figure 8 ijms-24-06981-f008:**
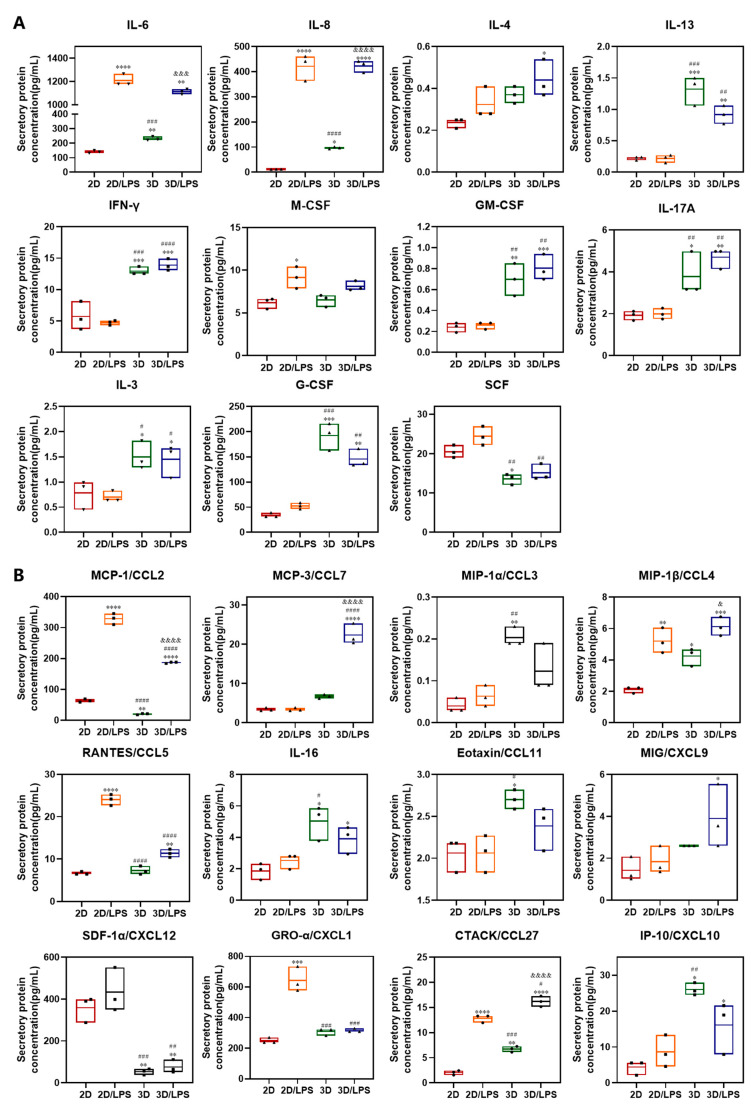
Immune cytokine assay of PDLSC-derived secretome after LPS and/or 3D culture treatment. (**A**) Cytokine expression involved in macrophage polarization and differentiation in 2D-, 2D/LPS-, 3D- and 3D/LPS-group-derived secretome; (**B**) Chemokine expression in 2D-, 2D/LPS-, 3D- and 3D/LPS-group-derived secretome. * *p* < 0.05, ** *p* < 0.01, *** *p* < 0.001, **** *p* < 0.0001, compared with 2D group; ^#^ *p* < 0.05, ^##^ *p* < 0.01, ^###^ *p* < 0.001, ^####^ *p* < 0.0001, compared with 2D/LPS group; ^&^
*p* < 0.05, ^&&&^
*p* < 0.001, ^&&&&^ *p* < 0.0001, compared with 3D group.

**Figure 9 ijms-24-06981-f009:**
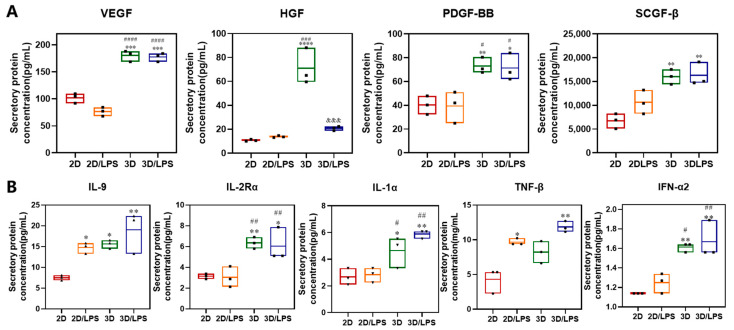
Immune cytokine assay of PDLSC-derived secretome after LPS and/or 3D culture treatment. (**A**) Growth factor expression in 2D-, 2D/LPS--, 3D and 3D/LPS-group-derived secretome. (**B**) Inflammatory cytokine expression in 2D-, 2D/LPS--, 3D and 3D/LPS-group-derived secretome. * *p* < 0.05, ** *p* < 0.01, *** *p* < 0.001, **** *p* < 0.0001, compared with 2D group; ^#^ *p* < 0.05, ^##^ *p* < 0.01, ^###^ *p* < 0.001, ^####^ *p* < 0.0001, compared with 2D/LPS group; ^&&&^ *p* < 0.001, compared with 3D group.

**Table 1 ijms-24-06981-t001:** Primer sequences for qRT-PCR.

Gene	Sequence	
*Homo GAPDH*	Forward	CCGCATCTTCTTTTGCGTCG
	Reverse	GGACTCCACGACGTACTCAG
*Homo IL-6*	Forward	ATGAACTCCTTCTCCACAAGCGC
	Reverse	GAAGAGCCCTCAGGCTGGACTG
*Homo IL-8*	Forward	ACACTGCGCCAACACAGAAATTA
	Reverse	TTTGCTTGAAGTTTCACTGGCATC
*Homo IL-10*	Forward	GATCTCCGAGATGCCTTCAG
	Reverse	ATCGATGACAGCGCCGTAGC
*Homo IDO*	Forward	GCCCTTCAAGTGTTTCACCAA
	Reverse	CCAGCCAGACAAATATATGCGA
*Homo TSG-6*	Forward	TGTCTGTGCTGCTGGATGGAT
	Reverse	TGTGGGTTGTAGCAATAGGCAT
*Homo TGF-β*	Forward	CTAATGGTGGAAACCCACAACG
	Reverse	TATCGCCAGGAATTGTTGCTG
*Mus GAPDH*	Forward	CACGACATACTCAGCACCAG
	Reverse	TCCAGTATGACTCTACCCAC
*Mus Arg-1*	Forward	ATGTCCCTAATGACAGCTCCT
	Reverse	GCTTCCAACTGCCAGACTGT
*Mus iNOS*	Forward	CCCTATTTCACCTGCAACAG
	Reverse	GCTTGTCCAGGGATTCTGG
*Mus IL-β*	Forward	TGCCACCTTTTGACAGTGATG
	Reverse	TGATGTGCTGCTGCGAGATT
*Mus TNF-α*	Forward	TAGCCCACGTCGTAGCAAAC
	Reverse	GCAGCCTTGTCCCTTGAAGA
*Mus IL-6*	Forward	TGATGGATGCTACCAAACTGGA
	Reverse	TGTGACTCCAGCTTATCTCTTGG
*Mus IL-10*	Forward	GCTGGACAACATACTGCTAACCG
	Reverse	CACAGGGGAGAAATCGATGACAG

## Data Availability

The data presented in this study are available on request from the corresponding authors.

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
