# Peer review of "Regulatory Effects of Three-Dimensional Cultured Lipopolysaccharide-Pretreated Periodontal Ligament Stem Cell-Derived Secretome on Macrophages"

_ijms, 2023, doi:10.3390/ijms24086981_

Round 1

Reviewer 1 Report

The present manuscript by Yuran et al highlights the use of the PDLSCs-SupraGel culture system to investigate the regulatory effect of PDLSCs-derived secretomes on macrophage polarization and migration via analysis of immune cytokines. This is a well-designed study, however, It seems that the author mixed the discussion section with the result section. It is important to follow the publisher’s guidelines and ensure that the results are presented and discussed separately.

and the Specific comments and suggestions are listed below.

1. Line 231 As described in the upper text, the Supragel was kept when you collect the sample. it would be helpful to discuss any potential effects of the SupraGel on protein concentration in the secretome.

2. Line 234, please specify which day of the secretome was used in this experiment.

3. In Line 243, it would be useful to clarify whether cells or supernatant were used in this experiment.

4. In Line 247, the conclusion could be made clearer with additional wording.

5. In the Materials and Methods section, please provide more specific details about which experiments used Method 3.4 and Method 3.6.

Author Response

General comment:

The present manuscript by Yuran et al highlights the use of the PDLSCs-SupraGel culture system to investigate the regulatory effect of PDLSCs-derived secretomes on macrophage polarization and migration via analysis of immune cytokines. This is a well-designed study, however, It seems that the author mixed the discussion section with the result section. It is important to follow the publisher’s guidelines and ensure that the results are presented and discussed separately.

Response: Thanks for the comments, we have distinguished the discussion section and the result sections as required.

Point 1: Line 231 As described in the upper text, the Supragel was kept when you collect the sample. it would be helpful to discuss any potential effects of the SupraGel on protein concentration in the secretome. 

Response 1: We thank the reviewer for pointing out this problem. On day 0, 3, 7 and 10, we collected the cell culture supernatant without Supragel and measured the protein concentrations. We have illustrated it in Section 3.4.

Point 2: Line 234, please specify which day of the secretome was used in this experiment.

Response 2: Thanks to the reviewer for pointing out this problem. The secretome used in detection of secreted protein concentration on day 0, 3, 7 and 10 was PDLSCs culture supernatants. The secretome we used in macrophages polarization, migration and the detection of immune cytokines was the mixture of cell culture supernatant on day 3 ,5, 7 and the secretome in Supragel collected at day 7. We explained it in section 3.4 and 3.6.

Point 3: In Line 243, it would be useful to clarify whether cells or supernatant were used in this experiment.

Response 3: Thank the reviewer for pointing this out. It is so important to show that we used the secretome instead of the cells in this experiment. We have modified it.

Point 4: In Line 247, the conclusion could be made clearer with additional wording.

Response 1: We thank the reviewer for the suggestion. We have modified it.

Point 5: In the Materials and Methods section, please provide more specific details about which experiments used Method 3.4 and Method 3.6.

Response 5: Thank the reviewer for the excellent advise. We have indicated which experiments used Methods 3.4 and 3.6.

Reviewer 2 Report

Reviewer comment

Title: Regulatory Effects of Three-Dimensional cultured Lipopolysac-2 charide Pretreated Periodontal Ligament Stem Cells-Derived 3 Secretome on Macrophages

In this paper, the authors aimed to investigate the role of MSSC-derived secretomes on regulation of the macrophage phenotypes. Few points that authors must address:

1.       Abstract: the abstract sentence structure is weird and hard to read. Recommend English proofing with professional writers. Recommend for abstract to follow narrative structures with the intro as well (macrophage is only mentioned at the second part, not much coverage on periondotal background in the abstract). Put in full name of PDLSC.

2.       Figure 1 – please cite you are using Biorender.com for the figure per their license agreement.

3.       Figure 2 – too small and too crowded data. Please split it into 2 figures.

4.       Figure 2C – what is the unit of relative mRNA level? Recommend use the term “fold-change”

5.       Figure 3C – you can show better by inverting the tubes.

6.       Figure 3E – scale bar missing. What are the green colour staining? Label please.

7.       Figure 3D – recommend higher magnification, the 3D cultures is too small.

8.       How did you determine the % of viability? Not in methods section.

9.       Line 318, figure 4B does not tally with the image.

10.   Figure 4B – use a higher magnification. Why is the live/dead totally black? Does that means the cells are dead?

11.   Figure 4C – how to you ensure the protein synthesis comparison is fair across groups? Pipetting the cells to control the cell number does not really help in ensuring consistent cell number. Recommend harvest and count the cells or perform PicoGreen to get the exact cell number. Then Consider diving doing protein amount/number of cells at the corresponding day to get a fair comparison.

12.   For all your RT-qPCR, when you say “relative to GapDH”, which set of experiment’s GapDH expression are you using?

13.   Figure 5 - I am not a big fan of comparing just concentrations in between set. It is not a totally fair comparison. But since it is a fairly large number of experiments, can the author add a small proof experiment to show that consistent cell number is plated with the protocols?

1.       Is there a reason you chose SupraGel? Will Matrigel work?

1.       Recommend the author to restructure the result and discussion section to bring out the significance of the data and work. As it is, it is hard to get the key novelty of the work. It might be easier to first compare LPS treated cultures with controls (I interpreting as key novelty) then compare 3D vs 2D cultures (this I think is less appealing at this stage as many groups has data to back this up, this set of data is more for validation and control studies).

Author Response

We are so grateful to the reviewers for pointing out these problems and excellent suggestions. 

Point 1: Abstract: the abstract sentence structure is weird and hard to read. Recommend English proofing with professional writers. Recommend for abstract to follow narrative structures with the intro as well (macrophage is only mentioned at the second part, not much coverage on periondotal background in the abstract). Put in full name of PDLSC.

Response 1: We have rewritten the abstract.

Point 2: Figure 1-please cite you are using Biorender.com for the figure per their license agreement.

Response 2: We have cited the figure was created with Biorender.com.

Point 3: Figure 2-too small and too crowded data. Please split it into 2 figures.

Response 3: We have splited it into 2 figures: Figure 2 and Figure 3.

Point 4: Figure 2C-what is the unit of relative mRNA level? Recommend use the term “fold-change”

Response 4: We have revised them in Figure 2, Figure 3 and Figure 7.

Point 5: Figure 3C-you can show better by inverting the tubes.

Response 5: We have inverted the tubes to verify the gelling in Figure 4C.

Point 6: Figure 3E-scale bar missing. What are the green colour staining? Label please.

Response 6: We have enlarged the pictures because they were too small to see the scale bar, and we have labeled the cells in green and red staining in Figure 5.

Point 7: Figure 3D-recommend higher magnification, the 3D cultures is too small.

Response 7: We have enlarged the pictures. Because higher magnification pictures showed few spheroids in one field of view.

Point 8: How did you determine the % of viability? Not in methods section.

Response 8: We have add the calculation of cells viability in section 3.5.

Point 9: Line 318, figure 4B does not tally with the image.

Point 10: Figure 4B-use a higher magnification. Why is the live/dead totally black? Does that means the cells are dead?

Response 9 and 10: We have replaced the images with higher magnification. After centrifugation, we separated all cells from the gel including dead and live cells, so the picture of supernatant after centrifugation did not show any cells.

Point 11: Figure 4C-how to you ensure the protein synthesis comparison is fair across groups? Pipetting the cells to control the cell number does not really help in ensuring consistent cell number. Recommend harvest and count the cells or perform PicoGreen to get the exact cell number. Then Consider diving doing protein amount/number of cells at the corresponding day to get a fair comparison.

Response 11: That is an important question. The purpose of this part of the study was to test the effect of LPS pretreatment and 3D culture on the protein secretion of PDLSCs under the same initial cell number, and to verify the effect of inflammatory stimulation and culture environment on the functional activity of PDLSCs. We tried to count the number of cells after 3D and 2D culture, but the cells after 3D culture were clustered and could not be counted accurately. PicoGreen is a suitable method, but since we regularly measure the secreted protein levels on 0, 3, 5, 7 and 10 and verifying this problem will require considerable additional experiments. According to the results of CCK8 in section 2.2, the cells proliferation of 2D culture was significantly higher than that of 3D culture within 7 days. If calculated based on protein quantity/cell number, the amount of protein secreted by one PDLSC in 3D culture would be much higher than that in 2D culture. Therefore, it was plausible to conclude that the secretory activity of PDLSCs was significantly promoted by 3D culture com-pared to 2D culture. We discussed this question in third paragraph of section 3. The comment and suggestion of the reviewer provided us with new ideas. We should measure protein secretion more accurately in order to obtain more direct results.

Point 12: For all your RT-qPCR, when you say “relative to GapDH”, which set of experiment’s GapDH expression are you using?

Response 12: We used each group's own GapDH.

Point 13: Figure 5-I am not a big fan of comparing just concentrations in between set. It is not a totally fair comparison. But since it is a fairly large number of experiments, can the author add a small proof experiment to show that consistent cell number is plated with the protocols?

Response 13: We performed standard cell counter plate counts before 2D and 3D cultures to ensure consistency in cell numbers, and CCK8 was performed immediately after cells inoculation, which showed no statistical difference between 2D and 3D cultures.

Point 14: Is there a reason you chose SupraGel? Will Matrigel work?

Response 14: Matrigel is also a kind of hydrogel that can be used for 3D cell culture. But it is composed of basement membrane extracts and naturally derived ECM constituents from animals  and presents drawbacks such as undefined ingredients, inconsistent stability, potential antigenicity and immunogenicity risks and it is usually expensive. But SupraGel can avoid these problems.

Point 15: Recommend the author to restructure the result and discussion section to bring out the significance of the data and work. As it is, it is hard to get the key novelty of the work. It might be easier to first compare LPS treated cultures with controls (I interpreting as key novelty) then compare 3D vs 2D cultures (this I think is less appealing at this stage as many groups has data to back this up, this set of data is more for validation and control studies).

Response 15: We have distinguished the discussion section and the result sections to bring out the significance of the data and work.

Round 2

Reviewer 1 Report

The paper has been improved and the answer in Auther's comments addressed my questions.

Author Response

General comment:

The paper has been improved and the answer in Auther's comments addressed my questions.

Response: Thank the reviewer for the comments to improve this paper.

Reviewer 2 Report

The authors addressed most of the comments. I just have one minor concern.

1. Line 562 - Are there any existing works that use this method of viability comparison? Grey values count the intensity across the pixels, which means the cell/spheroid's density per pixel will change this measurement. As such, it is hard to compare their viabilities fairly. How about use DAPI and cross stain with EthD-1 and use % viability = 1 - (EthD-1 count/DAPI count) instead?

Author Response

Point:  Line 562 - Are there any existing works that use this method of viability comparison? Grey values count the intensity across the pixels, which means the cell/spheroid's density per pixel will change this measurement. As such, it is hard to compare their viabilities fairly. How about use DAPI and cross stain with EthD-1 and use % viability = 1 - (EthD-1 count/DAPI count) instead?

Response: Thank the reviewer for the excellent advise. In 2D culture, live, dead or total cells (DAPI) could be counted to calculate the viability of PDLSCs, but in 3D culture, the cells appeared as clusters or spheroids, so it is impossible to count the cells. Therefore, the gray value of fluorescence intensity of live and dead cells spheroids was used to replace the number of cells in this study. Of course, this is also inaccurate as stated by the reviewer. We could qualitatively judge the viability of PDLSCs by laser confocal microscopy images, so we will remove Figure 5C if the reviewer considers it is inaccurate.